# Comparison of whole-body muscle imaging findings between GNE myopathy and other young adult-onset hereditary myopathies

Pattira Boonsri[1], Suppakorn Yamutai[2], Pramot Tanutit[1], Jirakit Sattayapornpipat[3], Chariyawan Charalsawadi[3], Prut Koonalintip[2], Pornchai Sathirapanya[2], Suwanna Setthawatcharawanich[2], Rattana Leelawattana[4], Pat Korathanakhun[2]*

1 Musculoskeletal Radiology Unit, Department of Radiology, Faculty of Medicine, Prince of Songkla University, Hatyai, Songkhla, Thailand, 2 Neurology Unit, Department of Internal Medicine, Faculty of Medicine, Prince of Songkla University, Hatyai, Songkhla, Thailand, 3 Genomic Excellence Center and Human Genetic Unit, Department of Pathology, Faculty of Medicine, Prince of Songkla University, Hat Yai, Songkhla, Thailand, 4 Endocrinology Unit, Department of Internal Medicine, Faculty of Medicine, Prince of Songkla University, Hatyai, Songkhla, Thailand

* patosk120@gmail.com

## Abstract

### Objectives

Previous muscle imaging studies of GNE myopathy are limited to the lower extremities. This study aimed to use whole-body MRI to differentiate between GNE myopathy and other young adult-onset hereditary myopathies.

### Materials and methods

This retrospective cohort study recruited patients with GNE myopathy or young adult-onset hereditary with limb girdle weakness pattern followed up in a single-center neuromuscular clinical registry between 2019 and 2023. Fatty tissue replacement was evaluated using a 5-point scale using T1-weighted images (T1WI) and proton-density fat fractions (PDFF) from mDIXON Quant images. Inflammation was evaluated using short tau inversion recovery imaging. The distribution and severity of muscle involvement in GNE myopathy were visualized using heat maps, and the parameters were tested for significance.

### Results

Of 103 patients, five with GNE myopathy and 10 with young adult-onset hereditary myopathy were recruited. Prominent fatty tissue replacement was seen in specific muscles with subtle active inflammation in GNE myopathy. The comparison of fatty tissue replacement between GNE and other young adult-onset hereditary myopathies exhibited the classic quadriceps sparing pattern in GNE myopathy group. Beyond these findings, latissimus dorsi showed the significantly lower fatty tissue

**Data availability statement:** All relevant data are within the paper and its Supporting Information files.

**Funding:** This study was funded by the Faculty of Medicine, Prince of Songkla University (REC. 66-539-14-3). The funders had no role in study design, data collection and analysis, decision to publish, or preparation of the manuscript.

**Competing interests:** The authors have declared that no competing interests exist.

replacement in the GNE group (median [IQR] of T1WI grade 1 [0, 1] vs. 3 [1, 3.4], $p = 0.04$) and mean (± S.D.) of PDFF in mDIXON Quant ($19.0 \pm 9.7$ vs. $42.6 \pm 22.7$, $p = 0.04$).

## Conclusion

The latissimus dorsi sparing out of proportion to periscapular weakness would be a novel differentiative feature of GNE myopathy.

---

## Introduction

GNE myopathy, an autosomal recessive hereditary myopathy, is caused by *GNE* gene mutations resulting in encoding errors that lead to defective sialylation of several myofiber proteins and subsequent muscle damage [1]. Besides the rarity of the disease, the age of onset in early adulthood mimicking other hereditary myopathies causes delayed diagnosis [1–3]. Although GNE myopathy had a recognizable clinical course showing early foot drop followed by proximal arms and legs weakness, the diagnosis of the GNE myopathy patients who presented with late stage of the disease is challenging as all muscles are affected and the clinical features mimics other hereditary myopathies. Muscle imaging is an non-invasive test that offer useful detail, particularly in deep muscles group, which are not easily detected in physical examinations. However, previous studies regarding the MRI in GNE myopathy were limited to descriptive methods and focused solely on the lower limbs with classic MRI sequences in T1-weighted (T1W) and short tau inversion recovery (STIR) images [4]. Therefore, this study aimed to identify the differentiating muscle involvement between GNE myopathy and other young adult-onset hereditary myopathy with limb girdle weakness pattern by using whole-body MRI.

## Materials and methods

### Patients and methods

This retrospective cohort study was approved by the ethics committee of the Faculty of Medicine, Prince of Songkla University (REC. 66-539-14-3), with the initial data access on 31st January 2024. The data from a neuromuscular clinical registry at a university-based hospital, a major referral center covering 14 provinces in southern Thailand, between 1st January 2019 and 31st December 2023 were retrospectively retrieved. The inclusion criteria were 1) diagnosis of GNE myopathy (confirmed by biallelic pathogenic variants of GNE) or young adult-onset hereditary myopathies with limb girdle weakness pattern (confirmed by either genetic tests, muscle biopsies, or MRI), and 2) available primary whole-body MRI imaging data.

### Data collection

Whole-body MRI was performed using a 1.5-T scanner (Philips Medical Systems, Best, Netherlands). The anatomic coverage extended from the skull base to both ankles and from the anterior to the posterior surface of the body in a head-first

direction. The anatomic data of the whole-body MRI parameters were classified into the six regions: 1) the cranium and neck, 2) shoulder girdle, 3) body and upper limbs, 4) pelvis, 5) thighs, and 6) lower legs. Three MRI sequences were performed with different purpose (Fig 1). T1WI aimed to detect fatty tissue replacement, classified into five grades [5] ranging between 0 (normal appearance) and 4 (end-stage muscle disease). STIR was used to determine edema or active inflammation, graded as present or absent. The mDIXON Quant technique quantified the percentage of fat replacement (PDFF). The MRI protocol, method of severity grading and landmark of measurement were provided in the supplementary data. MRI parameters were collected independently by two musculoskeletal radiologists (P.T. and P.B.), and any discordance in parameters was addressed through discussion until a consensus was reached.

The clinical data included demographic data, genetic test results, and functional scores evaluated within 1 month before or after MRI. The Brooke and Vignos scale [6] was used to assess upper and lower extremity function in both groups. Additionally, the GNE myopathy functional activity scale (GNEM-FAS) [7] was used as a disease-specific measurement in which the higher scores representing the better function.

### Data analysis

Descriptive demographic and clinical data are presented as numbers and percentages for discrete data and means ± standard deviations or medians (interquartile ranges) for continuous data. Detailed whole-body MRI findings from the GNE group are presented as a heat map indicating severity scores in each sequence. Whole-body MRI findings were compared between groups using the chi-squared test, independent t-test, or Mann–Whitney U test. Inter-rater reliability was examined using Cohen's kappa coefficient.

## Results

Of the 103 patients screened, 5 patients with GNE and 10 with other myopathies met the inclusion criteria and were enrolled.

### Clinical characteristics

All five patients with GNE myopathy exhibited classic clinical and pathological features. Two patients were siblings (patients 2 and 5); the other three were unrelated. The common missense variant of the *GNE* gene was present in one allele in all patients: (NM_001128227.3:c.2179G>A(p.Val696Met)). Patient 4 carried the same mutation in another allele, resulting in a homozygous variant. The other patients carried heterozygous variants in another allele; the missense variants in the other alleles were NM_001128227.3:c.1664C>T(p.Ala524Val) in the two siblings (patients 2 and 5) and NM_001128227.3:c.608G>A (p.Cys203Tyr) in the two unrelated patients (patients 1 and 3).

According to the GNEM-FAS, patients 1 and 2 had severely impaired function, while patient 4 had moderately impaired function; the remaining two patients had mildly impaired function.

Ten patients with young adult-onset hereditary myopathies with limb-girdle weakness pattern consisted of 4 genetically confirmed dystrophinopathy patients and 6 limb-girdle muscular dystrophy patients (2 genetically confirmed calpainopathy patients, 1 genetically confirmed dysferlinopathy and 3 patients with clinicopathological correlation with limb-girdle muscular dystrophy (LGMD) and having negative result for comprehensive neuromuscular gene panel test (211 genes testing provided by INVITAE company))

There was no significant difference in demographic data between the GNE and young adult-onset hereditary myopathies groups in terms of the number of men (3 [60.0%] vs. 6 [60.0%], respectively; $p$ = 1.000), mean age ± standard deviation (S.D.) at MRI (39.4 ± 10.2 years vs. 30.7 ± 7.4 years; t = 1.90, $p$ = 0.080), mean age ± S.D. at onset (27.4 ± 3.6 years vs. 21.7 ± 6.6 years; t = 1.77, $p$ = 0.100), and median duration (8.0 [7.0, 20.0] years vs. 5.0 [3.5, 12] years; $p$ = 0.462). However, the GNE myopathy group had a higher mean Brooke and Vignos scale score in the upper (4 ± 2.1 points vs. 1.8 ± 0.4 points, respectively; t = 3.27, $p$ = 0.006) and lower extremities (7.6 ± 1.3 points vs. 5.2 ± 2.4 points; t = 2.06, $p$ = 0.046) and a higher mean total Brooke and Vignos scale score (11.6 ± 3.3 points vs. 7.0 ± 2.6 points; t = 2.98, $p$ = 0.011).

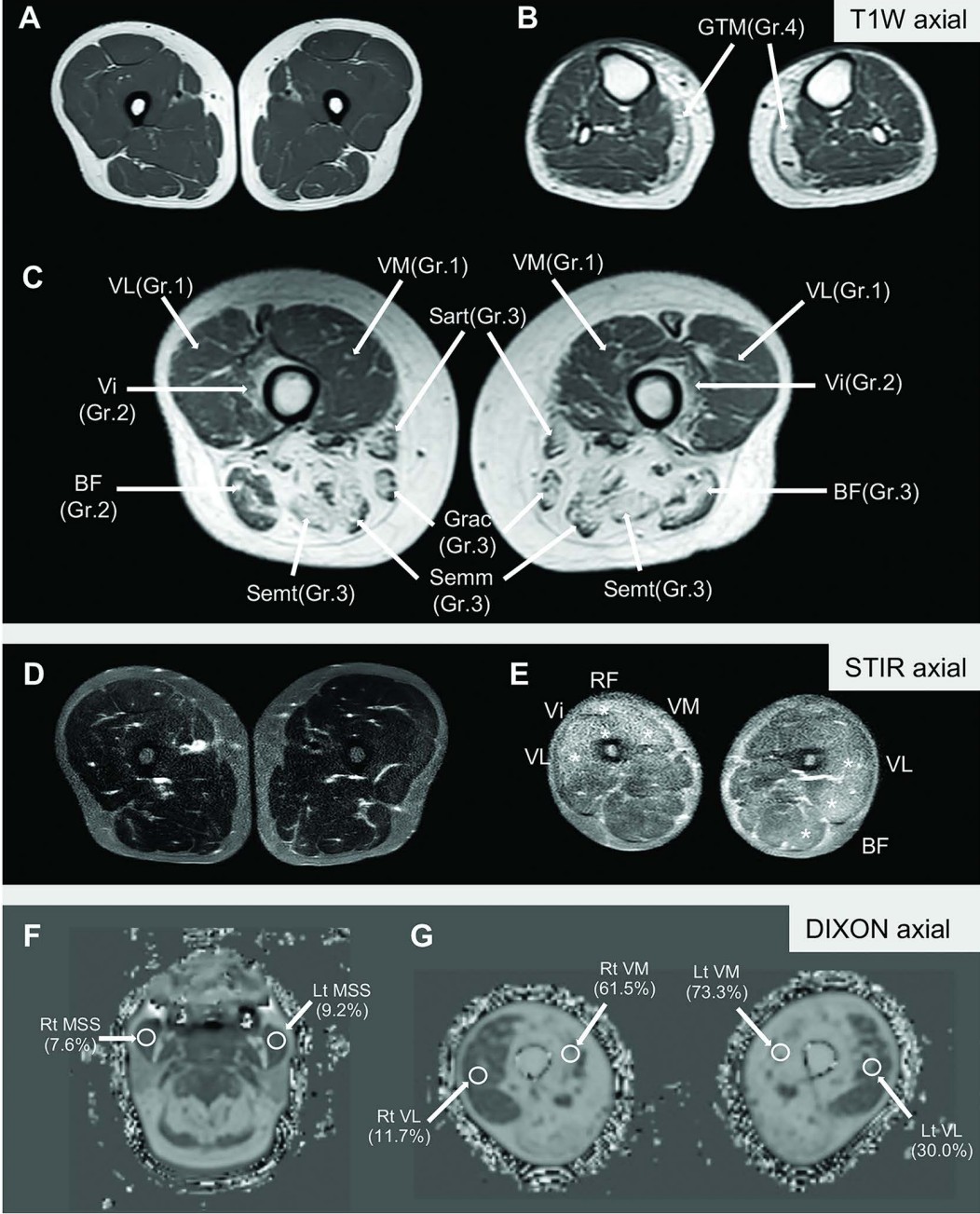

**Fig 1. Axial view of whole-body muscle magnetic resonance images for evaluating pathological changes in different sequences.** In the five grades of fatty tissue replacement on T1-weighted images **(A-C)**, grade 0 was considered the normal appearance **(A)**. Pathological changes of grade 1 (mild condition with trace amounts of fat signal (<25%)) were noted in the bilateral vastus medialis (VM) and vastus lateralis (VL) **(C)**, grade 2 (moderate condition with initial confluence observed in <50% of muscle) in the right biceps femoris (BF) long head and bilateral vastus intermedialis (Vi) **(C)**, grade 3 (severe condition with confluence observed in >50% of muscle) in sartorius (Sart), gracilis (Grac), semimembranosus (Semm), semitendinosus (Semt) both sides and left BF long head **(C)**, and grade 4 (end-stage disease with fat signals covering the entire muscle) in the bilateral gastrocnemius medial head (GTM) **(B)**. STIR images **(D-E)** revealed the presence of muscle edema (star marks) **(E)** compared with the absence of muscle edema **(D)**. Proton-density fat fractions (PDFF) evaluated on DIXON imaging **(F-G)** revealed a mild degree of fatty tissue replacement in bilateral masseter muscles, with PDFFs of 7.6% and 9.2% for the right and left muscles, respectively **(F)**. Thigh muscles showed a higher severity of fatty tissue replacement, with PDFFs of 11.7% and 30.0% for right and left vastus lateralis, respectively, as well as PDFFs of 61.5% and 73.3% for right and left vastus medialis, respectively **(G)**.

## Hallmark whole-body MRI characteristics of GNE myopathy

Among the three MRI sequences performed in this study, T1WI and mDIXON Quant sequences revealed obvious fatty tissue replacement with easily recognizable patterns. Although quantitative PDFF assessment in mDIXON Quant provided greater detail, the overall color maps of T1WI and mDIXON Quant were almost identical. In order to simplify the data visualization, we report the main characteristics of T1WI (Fig 2).

In the cranium and neck, the masticatory muscles (masseter, medial pterygoid, lateral pterygoid, and temporalis) were relatively spared on T1WI and the tongue was mildly affected in all patients, even in advanced stages. The neck muscles, namely the sternocleidomastoid and cervical extensors, were moderately to severely affected in advanced stages.

In the shoulder girdle, the periscapular muscles (supraspinatus, infraspinatus, and subscapularis) were affected in advanced stages, with greater severity in the subscapularis muscles. Additionally, the trapezius, deltoid, serratus anterior, and pectoralis muscles were affected in advanced stages, whereas the latissimus dorsi remained relatively spared even in advanced stages.

In the trunk and upper limb, the biceps were more severely affected than the triceps in all disease stages. In the forearms, the anterior compartment was more severely affected than the posterior and mobile-wad compartments. In the trunk, the psoas and spinal extensor muscles were primarily affected. In addition, lumbar extensors were more severely affected than thoracic extensors in all disease stages. Intercostal and abdominis muscles remained relatively spared even in advanced stages.

In the pelvic region, most muscles (gluteus maximus, gluteus medius, gluteus minimus, perineal muscles, adductor magnus, adductor longus, pectineus, adductor brevis, and iliotibial) were severely affected. However, the gluteus maximus and gluteus medius were relatively less affected than the gluteus minimus in early disease stages.

In the thigh, quadriceps sparing was seen. Although the quadriceps were affected in the advanced stages of the disease, the vastus lateralis remained relatively less affected than the vastus intermedialis, rectus femoris, and vastus medialis.

In the lower leg, all muscles (gastrocnemius medialis, gastrocnemius lateralis, soleus, tibialis anterior, tibialis posterior, extensor hallucis longus and extensor digitorum longus, flexor digitorum longus, and peronei) were affected in advanced disease stages. The soleus and tibialis anterior were severely affected in the early stages.

## Comparison of T1WI and mDIXON Quant imaging between GNE and young adult-onset hereditary myopathies group

As the findings in T1WI and mDIXON Quant imaging were almost identical, the comparison data were described together (Fig 2). Although the mDIXON Quant imaging provided the greater quantitiative detail, the technic for quantifying the PDFF were limited by the image resolution. Therefore, quantitative PDFF could not be accurately assessed in certain small muscles that could not clearly identify the muscles borders.

The comparison showed five muscle regions that differentiate GNE myopathy from the young adult-onset hereditary myopathies group. In the shoulder regions, latissimus dorsi in the GNE myopathy group had the lower median [IQR] fatty tissue replacement grade in T1WI (1 [0, 1] vs. 3 [1, 3.4], p = 0.04) and mean (± S.D.) PDFF in mDIXON Quant (19.0 ± 9.7 vs. 42.6 ± 22.7, p = 0.04) than those in the young adult-onset hereditary myopathies group.

In the trunk and upper limb regions, the GNE myopathy group demonstrated higher median [IQR] fatty tissue replacement grade in T1WI or mean (± S.D.) PDFF in mDIXON Quant than the young adult-onset hereditary myopathies group in anterior compartment of the forearm (T1WI grade 3 [2,3] vs. 1.8 [0, 2]; p = 0.04, PDFF 47.7 ± 27.9 vs. 20.2 ± 12.0, p = 0.04) and lumbar extensors (PDFF 71.5 ± 19.2 vs. 28.7 ± 17.2, p < 0.01).

In the pelvic region, GNE myopathy exhibited higher median [IQR] fatty tissue replacement grade in T1WI or mean (± S.D.) PDFF in mDIXON Quant in the gluteus minimus (T1WI grade 4 [4] vs. 3.2 [0.8, 4]; p = 0.04, PDFF 78.7 ± 6.3 vs. 41.2 ± 13.5, p = 0.03), adductor longus (T1WI grade 4 [4] vs. 3.2 [0.8, 4]; p = 0.04), and abdominal belt muscles (T1WI grade 4 [4] vs. 3 [0.5, 3.9]; p < 0.02) than the young adult-onset hereditary myopathies group.

**Fig 2. Heat map of the five grades of fatty tissue replacement on T1-weighted imaging of the five patients with GNE myopathy and the comparison of the fatty tissue replacement grades and PDFF between the GNE and non-GNE groups.**

Heat map showing characteristics of fatty replacement grade among GNE myopathy patients (Cases 1–5, Rt/Lt) in T1W images, with the following comparison data:

| Region | Muscles | T1W Median (IQR) GNE | T1W Non GNE | p-value | PDFF Mean (SD) GNE | PDFF Non GNE | p-value |
|---|---|---|---|---|---|---|---|
| Cranium | TMP | 0 (0,0) | 1 (0.2,1) | 0.09 | NA | NA | NA |
| | MSS | 0 (0,0) | 0 (0,0) | 0.35 | 5.1 (2.9) | 11.8 (14.5) | 0.33 |
| | MPt | 0 (0,0) | 0 (0,0.8) | 0.22 | NA | NA | NA |
| | LPt | 0 (0,0) | 0.5 (0,1) | 0.07 | 6.1 (5.4) | 6.2 (3.3) | 0.98 |
| | Tong | 1 (1,1) | 1 (1,2) | 0.30 | NA | NA | NA |
| | SCM | 0 (0,4) | 1 (0,2.2) | 0.56 | NA | NA | NA |
| | CExt | 1 (0,3) | 1 (0.2,1) | 0.70 | 17.6 (22.1) | 8.4 (4.2) | 0.21 |
| | Lcollis | 0 (0,0) | 0 (0,0) | 1.00 | NA | NA | NA |
| Shoulder | Ltsm | 1 (0,1) | 3 (1,3.4) | 0.04 | 19.0 (9.7) | 42.6 (22.7) | 0.04 |
| | Trpz | 1 (1,3) | 1 (0.2,1.4) | 0.61 | NA | NA | NA |
| | Delt | 2 (1,3) | 1.5(0.5,2.4) | 0.71 | 26.1 (23.4) | 19.9 (24.8) | 0.65 |
| | SSp | 0.5 (0.5,3) | 1.2(0.2,2.9) | 0.80 | NA | NA | NA |
| | ISp | 0.5 (0,2.5) | 1.5(0.2,2.8) | 0.66 | 28.9 (35.9) | 26.0 (26.6) | 0.86 |
| | SScp | 1.5 (1,3.5) | 2.5(0.5,3.8) | 0.85 | NA | NA | NA |
| | Pect | 0 (0,3) | 1.5 (0,2.4) | 1.00 | NA | NA | NA |
| | SA | 0 (0,3) | 3 (0.1,3.9) | 0.20 | 31.9 (30.3) | 43.3 (34.6) | 0.54 |
| Arm | BC | 2 (2,4) | 1.8 (0.2,3) | 0.53 | 48.9 (30.8) | 34.3 (28.4) | 0.38 |
| | TC | 2 (0,3) | 2 (0,2.4) | 0.70 | 38.2 (29.8) | 21.8 (23.2) | 0.26 |
| Forearm | Ant | 3 (2,3) | 1.8 (0,2) | 0.04 | 47.7 (27.9) | 20.2 (12.0) | 0.04 |
| | Mob | 1 (1,1.5) | 0.2 (0,1.8) | 0.29 | NA | NA | NA |
| | Post | 1.5 (1,2) | 0 (0,0) | 0.06 | 20.6 (17.6) | 12.0 (17.1) | 0.38 |
| Trunk | ICM | 1 (0,1) | 1 (0,1.8) | 0.60 | NA | NA | NA |
| | T.ext | 2 (1,3) | 1 (0.2,2) | 0.23 | 28.8 (14.1) | 27.2 (23.3) | 0.89 |
| | L.ext | 4 (3,4) | 3 (0.2,3) | 0.08 | 71.5 (19.2) | 28.7 (17.2) | <0.01 |
| | Psoas | 3 (2.5,4) | 1 (0.2,1.8) | 0.07 | 47.6 (34.0) | 29.4 (31.7) | 0.33 |
| | Abd | 2 (1,2) | 2 (0.2,2.4) | 0.90 | NA | NA | NA |
| Pelvis | GMax | 1 (1,3) | 3 (1,1.4) | 0.48 | 38.8 (37.9) | 48.6 (35.3) | 0.63 |
| | GMed | 3 (1.5,4) | 3.8 (0.2,4) | 0.90 | 49.0 (33.3) | 46.1 (35.9) | 0.88 |
| | Gmin | 4 (4,4) | 3.2 (0.8,4) | 0.04 | 78.7 (6.3) | 41.2 (13.5) | 0.03 |
| | AddM | 4 (4,4) | 3.2 (0.8,4) | 0.16 | NA | NA | NA |
| | AddL | 4 (4,4) | 3.2 (0.8,4) | 0.04 | NA | NA | NA |
| | Pec | 4 (4,4) | 3.5(0.5,3.9) | 0.18 | NA | NA | NA |
| | AdBlt | 4 (4,4) | 3 (0.5,3.9) | 0.02 | NA | NA | NA |
| Thigh | RF | 3.5 (3,4) | 3 (0.5,3) | 0.13 | 50.8 (31.7) | 33.6 (31.3) | 0.34 |
| | VL | 1 (1,2) | 3 (2.2,3.8) | 0.04 | 8.9 (5.6) | 43.7 (8.7) | 0.02 |
| | Vi | 3 (2,4) | 3 (3,4) | 0.95 | NA | NA | NA |
| | VM | 2 (1,3) | 3 (3,3) | 0.21 | 18.2 (27.1) | 49.7 (27.5) | 0.06 |
| | Sart | 3 (3,3.5) | 1 (0,1.9) | 0.03 | 60.0 (26.2) | 29.3 (16.1) | 0.04 |
| | Grac | 4 (4,4) | 1.5 (0,3) | <0.01 | 71.7 (9.0) | 29.8 (14.5) | <0.01 |
| | Semt | 4 (3,4) | 3.8 (0.8,4) | 0.60 | 75.3 (6.1) | 50.0 (36.5) | 0.15 |
| | Semm | 3 (3,4) | 3 (0.8,4) | 0.35 | 78.2 (6.1) | 48.8 (33.5) | 0.08 |
| | BFL | 3 (3,4) | 3 (0.8,3.8) | 0.56 | 65.2 (11.1) | 53.3 (33.4) | 0.46 |
| | BFS | 4 (4,4) | 1.5 (1,3.5) | 0.04 | 71.4 (8.9) | 39.0 (16.9) | 0.02 |
| Lower leg | GTM | 4 (4,4) | 2.5 (1,3) | 0.05 | 55.6 (26.0) | 43.9 (29.7) | 0.47 |
| | GTL | 4 (3,4) | 2 (1,3) | 0.15 | 51.3 (26.4) | 44.0 (29.1) | 0.64 |
| | Sol | 4 (4,4) | 2.5 (1.3,9) | 0.07 | 64.5 (27.1) | 39.1 (30.2) | 0.14 |
| | TA | 4 (3.5,4) | 2 (1.1,2.9) | 0.01 | 54.2 (24.2) | 29.9 (11.8) | 0.04 |
| | TP | 4 (4,4 | 1 (0,2) | 0.04 | 55.2 (32.1) | 20.8 (13.5) | 0.03 |
| | EH/D | 4 (4,4) | 2.5 (1.2,3) | 0.14 | NA | NA | NA |
| | FD | 4 (3,4) | 0.8 (0,2) | 0.03 | NA | NA | NA |
| | Peron | 3 (2,4) | 2.5(0.4,3.8) | 0.53 | 47.6 (32.0) | 36.7 (29.1) | 0.52 |

**GNE-FAS score:** Case 1 = 16, Case 2 = 11, Case 3 = 80, Case 4 = 48, Case 5 = 92

Colors / Fatty replacement (5 scales): 0 (green), 1 (yellow), 2 (orange), 3 (pink), 4 (purple)

The fatty tissue replacement severity was categorized into five grades, ranging from 0 (normal) to 4 (the most severe). Data are presented as medians and interquartile ranges. Abbreviations: TMP; temporalis, MSS; masseter, MPt; medial pterygoid, LPt lateral pterygoid, Tong; tongue, SCM; sternocleidomastoid, Cext; cervical extensor, Lcollis; longus collis, Ltsm; latissimus dorsi, Trpz; trapezius, Delt; deltoid, SSp; supraspinatus, Isp; infraspinatus, SScp; subscapularis, Pect; pectoralis minor and major, SA; serratus anterior, BC; biceps, TC; triceps, Ant; anterior compartment of the forearm, Mob; mobile-wad muscles of the forearm, Post; posterior compartment of the forearm, ICM; intercostal muscles, T.ext; thoracic extensor, L.ext; lumbar extensor, Psoas; psoas muscle, Abd; abdominal belt muscles, GMax; gluteus maximus, Gmed; gluteus medius, Gmin; gluteus minimus, AddM; adductor magnus, AddL; adductor longus, Pec; pectineus, RF; rectus femoris, VL; vastus lateralis, Vi; vastus intermedialis, VM; vastus medialis, Sart; sartorius, Grac; gracilis, Semt; semitendinosus, Semm;

semimembranous, BFL; biceps femoris long head, BFS; biceps femoris short head, GTM; gastrocnemius medialis, GTL; gastrocnemius lateralis, Sol; soleus, TA; tibialis anterior, TP; tibialis posterior, EH/D; extensor hallucis longus and extensor digitorum longus, FD; flexor digitorum longus and Peron; peronei.

In the thigh region, GNE myopathy had higher median [IQR] fatty tissue replacement grade in T1WI or mean (± S.D.) PDFF in mDIXON Quant in sartorius (T1WI grade 3 [3, 3.5] vs. 1 [0, 1.9], p = 0.03, PDFF 60.0 ± 26.2 vs. 29.3 ± 16.1, p = 0.04), gracilis (T1WI grade 4 [4] vs. 1.5 [0, 3], p < 0.01, PDFF 71.7 ± 9.0 vs. 29.8 ± 14.5, p < 0.01) and biceps femoris short head (T1WI grade 4 [4] vs. 1.5 [1, 3.5], p = 0.04, PDFF 71.4 ± 8.9 vs. 39.0 ± 16.9, p = 0.02) than the young adult-onset hereditary myopathies group. However, vastus lateralis in the GNE myopathy group had lower median [IQR] fatty tissue replacement grade in T1WI (1 [1,2] vs. 3 [2.2, 3.8], p = 0.04) and mean (± S.D.) PDFF in mDIXON Quant (8.9 ± 5.6 vs. 43.7 ± 8.7, p = 0.02) than those of the young adult-onset hereditary myopathies group, which emphasize the quadriceps sparing pattern in GNE myopathy.

In the lower leg, the GNE myopathy group demonstrated higher median [IQR] fatty tissue replacement grades or mean (± S.D.) PDFF in mDIXON Quant in tibialis anterior (T1WI grade 4 [3.5, 4] vs. 2 [1.1, 2.9]; p = 0.01, PDFF 54.2 ± 24.2 vs. 29.9 ± 11.8, p = 0.04), tibialis posterior (T1WI grade 4 [4] vs. 1 [0, 2]; p = 0.04, PDFF 55.2 ± 32.1 vs. 20.8 ± 13.5, p = 0.03), and flexor digitorum longus (T1WI grade 4 [3,4] vs. 0.8 [0, 2]; p = 0.03) than the young adult-onset hereditary myopathies group.

No significant differences were seen in fatty tissue replacement grades in the cranium and neck between the two groups.

## Comparison of STIR imaging between GNE and young adult-onset hereditary myopathies group

STIR imaging in patients with GNE myopathy revealed abnormalities in the latissimus dorsi, supraspinatus, infraspinatus, and vastus lateralis, which were shown to be relatively spared on T1WI. Comparing to the young adult onset hereditary myopathies group, GNE myopathy group showed the higher prevalence of biceps femoris short head muscle edema in STIR imaging (80.0% vs 10.0%, p = 0.02) (Fig 3).

The interrater reliability for the grading of whole-body MRI findings revealed good concordance between the two radiologists (T1WI, Kappa = 0.778, p < 0.001; STIR, Kappa = 0.668, p < 0.001; mDIXON Quant, Kappa = 0.750, p = 0.018).

## Discussion

Most muscle MRI studies on GNE myopathy are limited to the lower limbs, and a single descriptive study of the upper body region has been reported [1]. To our knowledge, this is the first study reporting whole-body MRI comparisons between GNE myopathy and young adult-onset hereditary myopathy. The result of this study not only emphasized the quadriceps sparing with anterior forearm muscle involvement in GNE myopathy, but also discovered the novel muscle sparing, namely latissimus dorsi, that potentially differentiate GNE myopathy from other young adult-onset hereditary myopathy.

### Genetic aspects of GNE myopathy in our cohort

Patients with GNE myopathy in our cohort indicated a genetic founder effect, as all patients carried the common missense pathogenic variant NM_001128227.3:c.2179G>A(p.Val696Met) in one allele. This is consistent with a previous report on the central Thai population and the other Asian populations [8–12]. On the other allele, the missense variant NM_001128227.3:c.1664C>T(p.Ala524Val) has been previously reported in Thai, Japanese, French, and Mexican populations [8,10,13,14]. In this cohort, two unrelated patients carried the novel missense variant NM_001128227.3:c.608G>A (p.Cys203Tyr), affecting the epimerase domain of the UDP-GlcNAc 2-epimerase/ManNAc kinase enzyme.

| Regions | Muscles | Case 1 Right | Case 1 Left | Case 2 Right | Case 2 Left | Case 3 Right | Case 3 Left | Case 4 Right | Case 4 Left | Case 5 Right | Case 5 Left | GNE | Non GNE | p-value |
|---|---|---|---|---|---|---|---|---|---|---|---|---|---|---|
| Cranium | TMP |  |  |  |  |  |  |  |  |  |  | 0 (0) | 0 (0) | 0.20 |
|  | MSS |  |  |  |  |  |  |  |  |  |  | 0 (0) | 0 (0) | 0.20 |
|  | MPt |  |  |  |  |  |  |  |  |  |  | 0 (0) | 3 (30.0) | 0.51 |
|  | LPt |  |  |  |  |  |  |  |  |  |  | 0 (0) | 4 (40.0) | 0.23 |
|  | Tong |  |  |  |  |  |  |  |  |  |  | 0 (0) | 1 (10.0) | 1.00 |
|  | SCM |  |  |  |  |  |  |  |  |  |  | 0 (0) | 0 (0) | 0.20 |
|  | CExt |  |  |  |  |  |  |  |  |  |  | 0 (0) | 0 (0) | 0.20 |
|  | Lcollis |  |  |  |  |  |  |  |  |  |  | 0 (0) | 0 (0) | 0.20 |
| Shoulder | Ltsm |  |  |  |  |  |  | ■ | ■ |  |  | 1 (20.0) | 1 (10.0) | 1.00 |
|  | Trpz | ■ | ■ |  |  |  |  |  |  |  |  | 1 (20.0) | 1 (10.0) | 1.00 |
|  | Delt |  |  |  |  |  |  |  |  |  |  | 0 (0) | 0 (0) | 0.20 |
|  | SSp | ■ | ■ | ■ | ■ |  |  |  | ■ |  |  | 3 (60.0) | 2 (20.0) | 0.25 |
|  | ISp | ■ | ■ | ■ |  |  |  | ■ | ■ |  |  | 3 (60.0) | 2 (20.0) | 0.25 |
|  | SScp |  |  |  |  |  |  | ■ | ■ |  |  | 1 (20.0) | 0 (0) | 0.33 |
|  | Pect |  |  |  |  |  |  |  |  |  |  | 0 (0) | 1 (10.0) | 1.00 |
|  | SA |  |  |  |  |  |  |  |  |  |  | 0 (0) | 0 (0) | 0.20 |
| Arm | BC |  |  |  |  |  |  |  |  |  |  | 0 (0) | 1 (10.0) | 1.00 |
|  | TC |  |  | ■ | ■ | ■ | ■ |  |  |  |  | 2 (40.0) | 1 (10.0) | 0.24 |
| Forearm | Ant |  |  |  |  |  |  |  |  |  |  | 0 (0) | 0 (0) | 0.20 |
|  | Mob |  |  |  |  |  |  |  |  |  |  | 0 (0) | 0 (0) | 0.20 |
|  | Post | ■ | ■ |  |  |  |  |  |  |  |  | 1 (20.0) | 1 (10.0) | 1.00 |
| Trunk | ICM |  |  |  |  |  |  |  |  |  |  | 0 (0) | 0 (0) | 0.20 |
|  | T.ext |  |  |  |  |  |  |  |  |  |  | 0 (0) | 0 (0) | 0.20 |
|  | L.ext |  |  |  |  |  |  |  |  |  |  | 0 (0) | 1 (10.0) | 1.00 |
|  | Psoas |  |  |  |  |  |  |  |  |  |  | 0 (0) | 0 (0) | 0.20 |
|  | Abd |  |  |  |  |  |  |  |  |  |  | 0 (0) | 0 (0) | 0.20 |
| Pelvis | GMax |  |  |  |  |  |  | ■ | ■ |  |  | 1 (20.0) | 1 (10.0) | 1.00 |
|  | GMed |  |  |  |  |  |  | ■ | ■ |  |  | 1 (20.0) | 1 (10.0) | 1.00 |
|  | Gmin |  |  |  |  |  |  |  |  |  |  | 0 (0) | 0 (0) | 0.20 |
|  | AddM |  |  |  |  |  |  |  |  |  |  | 0 (0) | 1 (10.0) | 1.00 |
|  | AddL |  |  |  |  |  |  |  |  |  |  | 0 (0) | 1 (10.0) | 1.00 |
|  | Pec |  |  |  |  |  |  |  |  |  |  | 0 (0) | 0 (0) | 0.20 |
|  | AdBlt |  |  |  |  |  |  |  |  |  |  | 0 (0) | 1 (10.0) | 1.00 |
| Thigh | RF |  |  |  |  |  |  |  |  |  |  | 0 (0) | 1 (10.0) | 1.00 |
|  | VL | ■ | ■ | ■ | ■ |  |  |  |  |  |  | 2 (40.0) | 2 (20.0) | 0.56 |
|  | Vi |  |  |  |  |  |  |  |  |  |  | 0 (0) | 1 (10.0) | 1.00 |
|  | VM |  |  | ■ | ■ |  |  |  |  | ■ | ■ | 2 (40.0) | 3 (30.0) | 1.00 |
|  | Sart |  |  |  |  |  |  |  |  |  |  | 0 (0) | 0 (0) | 0.20 |
|  | Grac |  |  |  |  |  |  |  |  |  |  | 0 (0) | 0 (0) | 0.20 |
|  | Semt |  |  |  |  |  |  |  |  |  |  | 0 (0) | 0 (0) | 0.20 |
|  | Semm |  |  |  |  |  |  | ■ | ■ |  |  | 1 (20.0) | 0 (0) | 0.33 |
|  | BFL |  |  |  |  |  |  | ■ | ■ | ■ | ■ | 2 (40.0) | 1 (10.0) | 0.24 |
|  | BFS | ■ | ■ | ■ | ■ |  |  | ■ | ■ | ■ | ■ | 4 (80) | 1 (10.0) | 0.02 |
| Lower leg | GTM |  |  |  |  | ■ | ■ | ■ | ■ |  |  | 2 (40.0) | 3 (30.0) | 1.00 |
|  | GTL |  |  |  |  |  |  |  |  |  |  | 0 (0) | 2 (20.0) | 0.52 |
|  | Sol | ■ | ■ |  |  |  |  |  |  |  |  | 1 (20.0) | 1 (10.0) | 1.00 |
|  | TA |  |  | ■ | ■ |  |  | ■ | ■ |  |  | 2 (40.0) | 4 (40.0) | 1.00 |
|  | TP |  |  | ■ | ■ |  |  |  | ■ |  |  | 2 (40.0) | 4 (40.0) | 1.00 |
|  | EH/D |  |  |  |  |  |  |  |  |  |  | 0 (0) | 2 (20.0) | 0.52 |
|  | FD |  |  |  |  |  |  | ■ | ■ |  |  | 1 (20.0) | 1 (10) | 1.00 |
|  | Peron |  |  |  |  |  |  | ■ | ■ |  |  | 1 (20.0) | 0 (0) | 0.33 |

**Fig 3. Heat map of the presence of muscle edema on STRI imaging in the five patients with GNE myopathy and the comparison of the prevalence of muscle edema between the GNE and non-GNE groups.** Edematous muscles are indicated in red. Data are presented as medians and interquartile ranges. The same abbreviations in Fig 2 were used.

## Clinical characteristics and whole-body MRI in GNE myopathy

**The practical use of T1WI, mDIXON Quant and STIR.** MRI protocols in previous studies were limited to T1WI and STIR. The current study used mDIXON Quant imaging in addition to standard T1WI and STIR. Herein, hallmark MRI characteristics were observed on T1WI and mDIXON Quant image which were identical. The fatty tissue replacement

grading in T1WI is a simple visual scale method and offers a reliable result confirm by good interrater reliability in our study. The detail of the fatty tissue replacement grade was sufficient for clinical practice. Although the mDIXON Quant image provide the greater quantitative detail, several technical limitations were considered. First, this method is labor and time consuming. Second, the PDFF was measure in the focal area specified in the standardized axial cut of each muscle. With this technic, the sampling error might occur in the muscle that had scattered patchy fatty tissue replacement while the visual scale grading in T1WI evaluate the whole muscle mass. Third, due to the limited mDIXON Quant image resolution, the accuracy of muscle border identification, especially in the small muscle, was inconsistent which increase the possibility of data collection error. To our experience, we would suggest that the visual scale grading in T1WI was reliable method and offer sufficient data for clinical practice, while the quantitative PDFF in mDIXON Quant image was suitable for the research purpose.

Interestingly, muscle edema in STIR sequences were usually observed in muscles exhibiting mild-to-moderate fatty tissue replacement in T1WI and mDIXON Quant. This might have emphasized certain muscle groups that were actively inflamed in each disease stage. Herein, vastus lateralis muscle edema were observed in the advanced stage GNE myopathy patient. This is consistent with the sequential involvement of quadriceps in which the vastus lateralis is affected later than the vastus medialis, vastus intermedialis, and rectus femoris. In addition, in the shoulder girdle, STIR revealed pathological changes in the supraspinatus and infraspinatus muscles in advanced GNE myopathy.

**The novel data from whole-body MRI in GNE myopathy.** In general, our study emphasized the hallmark of GNE myopathy including quadriceps sparing pattern, especially vastus lateralis, and specifically involvement in anterior forearm, biceps femoris short head and anterior lower leg muscles. Using the whole-body MRI, this study revealed additional interesting findings in certain upper body parts.

In the cranium and neck, the tongue was mildly affected, with grade 1 fatty tissue replacement in all patients. This contrasts with a study in the Italian population showing tongue muscle sparing even in advanced stages of the disease [1]. Conversely, herein, the cervical extensor muscles were severely affected in advanced stages, while they were spared in the Italian study [1].

In the shoulder girdle, the trapezius, deltoid, serratus anterior, pectoralis, and periscapular muscles (supraspinatus, infraspinatus, and subscapularis) exhibited late involvement in advanced stages, with greater disease severity observed in the subscapularis muscles. This is consistent with the findings of the Italian study [1]. However, we discovered that latissimus dorsi was relative spared even the other shoulder girdle muscles were severely affected.

In the trunk and upper limbs, significantly higher fatty tissue replacement grades were observed on T1WI in lumbar extensors in the GNE myopathy group. Patients with GNE myopathy showed early involvement of the psoas and spinal extensors, with greater severity in lumbar extensors than in thoracic extensors. In the rare GNE case with specific mutation, isolated lumbar paraspinal muscle atrophy may occur while the psoas muscles were preserved [15].

**Latissimus dorsi muscle sparing as a potential novel differentiative features between GNE myopathy and young adult-onset hereditary myopathies with limb-girdle weakness pattern.** Apart from the hallmark feature of GNE, this study found the significant less involvement of latissimus dorsi than those in the young adult-onset hereditary myopathies with limb-girdle weakness pattern group. The comparing group in this study were comprised of LGMD and dystrophinopathy.

LGMD is a collective group sharing the core clinical feature with heterogenous causative gene. Few studies had focused on the whole-body MRI in the LGMD. Among the LGMDR1 (calpainopathy), LGMDR2 (dysferlinopathy) and LGMDR3−6 (sarcoglycanopathy), the degree of latissimus dorsi involvement were correlated to the severity of periscapular and shoulder girdle muscles which was in contrast to our results [16–21]. In LGMDR12 (anoctamin-5), not only the latissimus dorsi were spared, but all of the periscapular and shoulder girdle muscle were also spared [22].

The study regarding the upper limbs muscle MRI in dystrophinopathies were limited in number and the shoulder region MRI were rarely studied. One study had investigated the becker muscular dystrophy patients and found the involvement

mainly in biceps, triceps and periscapular muscle with deltoid sparing. Unfortunately, this study did not include latissimus dorsi in the data collection [23]. From our result, it is possible that the relatively sparing latissimus dorsi out of proportionated to the severity of periscapular muscle would be a differentiative feature of GNE myopathy.

Another possible suggestive feature of GNE from our data was lumbar extensor involvement. This feature might be not specific to GNE as the lumbar extensor involvement were also found in LGMD and dystrophinopathies, but our data showed significantly more severe in GNE than those of compared group [16,24,25]. Unfortunately, we did not perform the individual lumbar extensor muscle, namely lumbar erector spinae and quadratus lumborum due to the limited MRI image resolution.

### Strengths, limitations, clinical implication and suggestions

To the best of our knowledge, this is the first study to investigate the characteristics of patients with GNE myopathy using whole-body MRI. In addition, this study's comparative design supports the applicability of our findings. Another strength of this study was the systematic grading of abnormalities in each MRI sequence, which was objectively reproducible. Although MRI evaluation using visual scores may be operator-dependent, our study controlled for this potential bias by having two musculoskeletal radiologists assess MRI findings independently, with good inter-rater reliability. Furthermore, mDIXON Quant imaging was performed in addition to the conventional T1WI and STIR sequences.

As GNE myopathy is rare, the sample size was inevitably limited. Despite the retrospective design of our study, we minimized missing data by using the neuromuscular clinic registry. However, the common genetic founder effect in our study may limit the generalizability. Although the mDIXON Quant technique provide good quantitative detail, its practical use was difficult because of several challenges. First, the accurate borders of the small muscle group could not be identified clearly owing to the limited resolution. Second, PDFF measurement in some muscles might have been underestimated because the standardized landmarks for the measurement area might have been spared in some scatterable fatty tissue-infiltrating muscles [26]. Third, PDFF measurement is time-consuming. We considered the five-grade fatty tissue replacement scale using T1WI provide sufficient and reliable data with a simple practical workflow. There are some concerns for using this research data in the clinical practice. In general approach, the clinician always start the clinical approach of hereditary myopathy by the distribution of the affected part in which GNE myopathy started with distal myopathy. However, in case of late presentation with advanced disease stage, the affected parts blended both proximal and distal muscle. The finding in this research would guide some clinical clues to recognize GNE myopathy in case of late presentation as the population in this study were in the moderate to advanced disease stage. The comparion study of early GNE myopathy with other distal myopathy will be the area of future interest. In addition, muscle imaging research in GNE myopathy should expand the imaging protocol to include more body regions.

### Conclusion

Beyond the classic quadriceps sparing with anterior forearm, anterior lower leg and biceps femoris short head involvement, the latissimus dorsi sparing out of proportion to periscapular weakness would be a novel differentiative feature of GNE myopathy. For clinical practice, the five-grade fatty tissue replacement scale using T1WI provide sufficient and reliable data.

### Supporting information

**S1 File. Raw data of GNE cohort.**
(XLSX)

**S2 File. Supplementary materials.**
(DOCX)

## Acknowledgments

We thank Dr. Ponlagrit Kumwichar and Mr. Sarawut Sukkhumfor their help regarding the statistical analyses.

## Author contributions

**Conceptualization:** Pattira Boonsri, Suppakorn Yamutai, Pramot Tanutit, Jirakit Sattayapornpipat, Chariyawan Charalsawadi, Prut Koonalintip, Pornchai Sathirapanya, Suwanna Setthawatcharawanich, Rattana Leelawattana, Pat Korathanakhun.

**Data curation:** Pattira Boonsri, Suppakorn Yamutai, Pramot Tanutit, Chariyawan Charalsawadi, Pat Korathanakhun.

**Formal analysis:** Pattira Boonsri, Suppakorn Yamutai, Pramot Tanutit, Prut Koonalintip, Pat Korathanakhun.

**Funding acquisition:** Pat Korathanakhun.

**Investigation:** Pattira Boonsri, Suppakorn Yamutai, Pat Korathanakhun.

**Methodology:** Pattira Boonsri, Pat Korathanakhun.

**Project administration:** Pat Korathanakhun.

**Resources:** Pattira Boonsri, Pramot Tanutit, Jirakit Sattayapornpipat, Chariyawan Charalsawadi, Pat Korathanakhun.

**Supervision:** Pramot Tanutit, Chariyawan Charalsawadi, Pornchai Sathirapanya, Suwanna Setthawatcharawanich, Rattana Leelawattana, Pat Korathanakhun.

**Validation:** Pat Korathanakhun.

**Visualization:** Pattira Boonsri, Suppakorn Yamutai, Pramot Tanutit, Jirakit Sattayapornpipat, Prut Koonalintip, Pornchai Sathirapanya, Rattana Leelawattana, Pat Korathanakhun.

**Writing – original draft:** Pattira Boonsri, Suppakorn Yamutai, Pramot Tanutit, Jirakit Sattayapornpipat, Chariyawan Charalsawadi, Prut Koonalintip, Pornchai Sathirapanya, Suwanna Setthawatcharawanich, Rattana Leelawattana, Pat Korathanakhun.

**Writing – review & editing:** Pattira Boonsri, Suppakorn Yamutai, Pramot Tanutit, Jirakit Sattayapornpipat, Chariyawan Charalsawadi, Prut Koonalintip, Pornchai Sathirapanya, Suwanna Setthawatcharawanich, Rattana Leelawattana, Pat Korathanakhun.

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
