## [Decision Letter · Decision Letter 0]

17 Jun 2025

Dear Dr. Korathanakhun,

Thank you for submitting your man uscript to PLOS ONE. After careful consideration, we feel that it has merit but does not fully meet PLOS ONE’s publication criteria as it currently stands. Therefore, we invite you to submit a revised version of the manuscript that addresses the points raised during the review process.

We look forward to receiving your revised manuscript.

Kind regards,

Mainak Bardhan, MD

Academic Editor

PLOS ONE

Journal Requirements:

[The Faculty of Medicine, Prince of Songkla University (REC. 66-539-14-3)].

Reviewers' comments:

Reviewer's Responses to Questions

**Comments to the Author**

1. Is the manuscript technically sound, and do the data support the conclusions?

Reviewer #1: Yes

Reviewer #2: Partly

2. Has the statistical analysis been performed appropriately and rigorously?

Reviewer #1: Yes

Reviewer #2: I Don't Know

3. Have the authors made all data underlying the findings in their manuscript fully available?

Reviewer #1: Yes

Reviewer #2: No

4. Is the manuscript presented in an intelligible fashion and written in standard English?

Reviewer #1: Yes

Reviewer #2: Yes

Reviewer #1: The article is well written and the description of this rare entity is a valuable contribution to the scientific community.

However, the article is a little too long, and in my opinion, some parts should be removed to make it easier to read and avoid drowning out the main message:

- I don't believe that quantitative sequences are necessary

- in view of the small number of myopathy cases I've noted typed for comparison, I don't think their description is necessary either a single fsh is not relevant for comparison with the GNE myopathy panel

- a single descriptive analysis of your population already seems very relevant to me

- to improve overall reading, a diagram with a color scale representing the frequency of muscles affected by anatomical region could be suggested.

Reviewer #2: The novelty of this study lies in the whole body protocol, evaluating cranial and upper limb muscles, and in the partial homogeneity of the population (the common c.2179G>A variant). Nevertheless, I do not understand how the comparison group adds anything to the study and find numerous inaccuracies and redundant concepts.

The author claims that the inclusion criteria for the "control" group are young adult-onset hereditary myopathies (confirmed through genetic tests, muscle biopsies, or MRI). It is unclear whether some of the other myopathies used as controls have a genetic diagnosis. (e.g. generic limb-girdle muscular dystrophy?). If they do not, it must be indicated how the diagnosis of GNE has been excluded. Calpainopathy is an LGMD, why is it listed separately? It would also be useful to specify why those myopathies were chosen: typically, GNE has a foot drop onset (This is also acknowledged in the author’s discussion (p. 18 line 398)) and many other myopathies (e.g. distal early onset myopathies such as MYH7, TTN) are not considered.

Many data are similar between T1 and DIXON image analysis, so it is preferrable to eliminate repetitions and focus on muscle groups where DIXON images are more accurate for their best resolution. In fact, in the discussion paragraph "Variations in findings across MRI sequences" there is absolutely no comparison between the two techniques and it is not clear why I should prefer either one or the other.

In lower limb analysis, it is well known that Biceps femoris short head is the first muscle involved in GNE and is therefore configured as a pathognomic for diagnosis along with quadriceps sparing. By not differentiating the two ends of the femoral biceps in the analysis, the author loses an important muscle, if not the pivotal muscle, able to differentiate this pathology from the others included. It is indeed surprising that a difference was NOT identified between the two groups in the thigh section.

":Herein, in patients with GNE with GNEM-FAS scores indicative of more severe disease, pathological changes were seen in advanced stages in the vastus lateralis on STIR imaging". This sentence is not supported by the data reported in the results

p 17 line 389-391 the author should comment on the results of his study and comparisons with the "small" control group e.g., the obturator internus which is mentioned here, does not appear among the muscles analyzed by the author nor among those significant. The comparison with FSHD (there is only one FSHD in the comparison group) or myofibrillar myopathies (they are not present in the comparison group) is mentioned several times in the discussion. The differential diagnosis with LGMD that would have led the author to include them is never addressed.

p 17 line 392, ST involvement is typical of ONE myofibrillar myopathy = desminopathy, and also an early feature of HMERF (10.1016/j.nmd.2017.12.002)

In all this discussion It’s not clear what the author means with "the correlation between disease severity and certain muscles"

Other comments:

please use either QUANT or quant in the text

pag 6 line 122 PDFF should de spelled also in the text not only in the figure

Avoid using p-value when the finding is not significant.

p 17 lines 362-363 a reference seems missing while reference 26 is unnecessary

p 17 lined 369-371: this phrase is in contrast with the results of the study and not clear

**Do you want your identity to be public for this peer review?** For information about this choice, including consent withdrawal, please see our Privacy Policy

Reviewer #1: **Yes:**  Marie FARUCH BILFELD

Reviewer #2: No

---

## [Author Response · Author response to Decision Letter 1]

25 Jul 2025

Dear Editor and reviewers,

Thank you for your valuable comments that sharpening our idea. I have thoroughly read the reviewers’ comments and edited the manuscript. The major comment effects the major change in the control group that showed heterogenous clinical and genetic result. According to the clinical practice, clinicians usually start with recognizing the distribution of muscle weakness. Therefore, we reset the control group by recruiting “young adult-onset hereditary myopathy with limb-girdle weakness pattern”.

We have explained the reason why the control group was selected in the introduction section as the following “Although GNE myopathy had a recognizable clinical course showing early foot drop followed by proximal arms and legs weakness, some patients, especially in case of longstanding inconclusive diagnosis, may present with late stage which all muscle were involved and the patient may not recognize the sequential symptoms accurately. In this setting, the differentiation GNE from other limb girdle pattern myopathy was very difficult and additional investigations are needed.”

Considering the article length, we optimized this issue by concision the method and refer all details to the supplementary data. We also merge the comparison data of T1WI grading and PDFF into a single figure (figure 2). In addition, the same abbreviations used in the figure 3 legend were shortening by referring to the abbreviation in the figure 2.

Due to the change of the control group, the results, figures and discussion were all edited. Please find the response to reviewers’ comments in the separate file.

Sincerely Yours,

Pat Korathanakhun, M.D.

---

## [Decision Letter · Decision Letter 1]

14 Sep 2025

Dear Dr. Korathanakhun,

Thank you for submitting your manuscript to PLOS ONE. After careful consideration, we feel that it has merit but does not fully meet PLOS ONE’s publication criteria as it currently stands. Therefore, we invite you to submit a revised version of the manuscript that addresses the points raised during the review process.

We look forward to receiving your revised manuscript.

Kind regards,

Mainak Bardhan, MD

Academic Editor

PLOS ONE

Journal Requirements:

Additional Editor Comments:

Reviewer #2: Please revise the type setting errors and other points as suggested

Reviewers' comments:

Reviewer's Responses to Questions

**Comments to the Author**

Reviewer #2: All comments have been addressed

2. Is the manuscript technically sound, and do the data support the conclusions?

Reviewer #2: Yes

3. Has the statistical analysis been performed appropriately and rigorously?

Reviewer #2: I Don't Know

4. Have the authors made all data underlying the findings in their manuscript fully available?

Reviewer #2: Yes

5. Is the manuscript presented in an intelligible fashion and written in standard English?

Reviewer #2: No

Reviewer #2: The authors made significant improvement to the manuscript that now sums up the main finding.

However I suggest to double check the English translation in the tracked change text as some concept are not clear e.g. pg.3 “ some patients, especially in case of longstanding inconclusive diagnosis, may present with late stage which all muscle were involved and the patient may not recognize the sequential symptoms accurately”

In the abstract, the conclusion are not supported by the results (anterior forearm, biceps femoris involvement is not enumerated in the results section of the abstract. Moreover, anterior leg involvement is not even a finding of this study as “The soleus and tibialis anterior were severely affected in the early stages” (pg 14 and pg 25 line 525)

minor comment - vastus intermedius instead of intermedialis (e.g. Fig 1)

please replace decimal commas with point throughout the text (e.g. pg 15)

pg 23 line 488 typo “several ltechnical imitations were considered”—> several technical limitations…

pg 26 line 547 and pg 26 line 566 typo “LGDM” —> LGMD

pg 26 line 551 typo “LGMDR36” —> LGMDR3-6

**Do you want your identity to be public for this peer review?** For information about this choice, including consent withdrawal, please see our Privacy Policy

Reviewer #2: No

---

## [Author Response · Author response to Decision Letter 2]

17 Sep 2025

Dear reviewers,

We wish to re-submit the manuscript titled “Comparison of whole-body muscle imaging findings in GNE myopathy and other young adult-onset hereditary myopathies” The manuscript has been rechecked and appropriate changes have been made in accordance with the reviewers’ suggestions. The responses to their comments have been prepared and attached herewith.

We thank you and the reviewers for your thoughtful suggestions and insights, which have enriched the manuscript and produced a better and more balanced account of the research. We hope that the revised manuscript is now suitable for publication in your journal.

Sincerely,

Pat Korathanakhun, M.D.

Associate Professor in Neurology, Faculty of Medicine, Prince of Songkla University

15 Kanjanavanich Road, Hat Yai, Songkhla 90110, Thailand

Tel: +66 7-4451452

Fax: +66 74-429385

Email: patosk120@gmail.com

---

## [Decision Letter · Decision Letter 2]

2 Jan 2026

Comparison of whole-body muscle imaging findings between GNE myopathy and other young adult-onset hereditary myopathies

PONE-D-25-08873R2

Dear Dr. Korathanakhun,

We’re pleased to inform you that your manuscript has been judged scientifically suitable for publication and will be formally accepted for publication once it meets all outstanding technical requirements.

Kind regards,

Vinay Kumar, Ph.D.

Academic Editor

PLOS One

Additional Editor Comments (optional):

Reviewers' comments:

Reviewer's Responses to Questions

**Comments to the Author**

Reviewer #2: All comments have been addressed

Reviewer #3: (No Response)

2. Is the manuscript technically sound, and do the data support the conclusions?

Reviewer #2: Yes

Reviewer #3: No

3. Has the statistical analysis been performed appropriately and rigorously?

Reviewer #2: Yes

Reviewer #3: N/A

4. Have the authors made all data underlying the findings in their manuscript fully available?

Reviewer #2: Yes

Reviewer #3: Yes

5. Is the manuscript presented in an intelligible fashion and written in standard English?

Reviewer #2: Yes

Reviewer #3: Yes

Reviewer #2: The final version of the manuscript has improved. Please carefully review minor typos as STIR spelling in Fig 3 legend

Reviewer #3: This study aimed to use whole-body MRI to differentiate between GNE myopathy and other young adult-onset hereditary myopathies.

The retrospective study considered a significant number of patients MRI with a technically sound methodology.

However out of 103 patients, file only with GNE myopathy and 10 with young adult-onset hereditary myopathy were included.

Unfortunately I regret to notice that either the premises or the conclusion of the manuscript contain incorrect statements.

In fact, it is reported that:

“Previous muscle imaging studies of GNE myopathy are limited to the lower

extremities.” This is not correct (see Torchia E. et al.)

The main conclusion again is not correct:

“The latissimus dorsi sparing out of proportion to periscapular weakness

would be a novel differentiative feature of GNE myopathy”

This has been already shown and published by Torchia E. et al.

The use of the mDIXON Quant technique to quantify the percentage of fat replacement

is basically the only new contribution.

Some of the reported features in the comparison between LGMD and GNE myopathy may be interesting clues, but after all the limited size of the analysed sample makes difficult the generalizability of these observations.

**Do you want your identity to be public for this peer review?** For information about this choice, including consent withdrawal, please see our Privacy Policy

Reviewer #2: No

Reviewer #3: No

---

## [Editor Report · Acceptance letter]

PONE-D-25-08873R2

PLOS One

Dear Dr. Korathanakhun,

I'm pleased to inform you that your manuscript has been deemed suitable for publication in PLOS One. Congratulations! Your manuscript is now being handed over to our production team.

Kind regards,

on behalf of

Dr. Vinay Kumar

Academic Editor

PLOS One